# Genetic Characterization of Kazakhstan Isolates: Avian Influenza H9N2 Viruses Demonstrate Their Potential to Infect Mammals

**DOI:** 10.3390/v17050685

**Published:** 2025-05-08

**Authors:** Barshagul Baikara, Kobey Karamendin, Yermukhammet Kassymbekov, Klara Daulbayeva, Temirlan Sabyrzhan, Sardor Nuralibekov, Yelizaveta Khan, Nurlan Sandybayev, Sasan Fereidouni, Aidyn Kydyrmanov

**Affiliations:** 1Research and Production Center for Microbiology and Virology, Almaty A26T6C0, Kazakhstan; 2Kazakhstan-Japan Innovation Centre, Kazakh National Agrarian Research University (KazNARU), Almaty 050000, Kazakhstan; 3Research Institute of Wildlife Ecology, University of Veterinary Medicine Vienna, 1160 Vienna, Austria

**Keywords:** influenza A virus, H9N2 subtype, bird, genome, variability, phylogenesis, hemagglutinin, adaptation, transmission

## Abstract

Low pathogenic H9N2 avian influenza viruses have become widespread in wild birds and poultry worldwide, raising concerns about their potential to spark pandemics or their role in enhancing the virulence and infectivity of H5Nx viruses through genetic reassortment. Therefore, influenza monitoring studies, including those of H9N2 viruses, are crucial for understanding, evaluating, and mitigating the risks associated with avian infections, and have broader implications for global public health. Although H9N2 viruses are not considered enzootic in Kazakhstan, they have been repeatedly detected in wild waterfowls and domestic poultry. In this study, all eight gene segments of influenza A/H9N2 viruses isolated in various regions of Kazakhstan between 2014 and 2020 were sequenced and analyzed. Molecular characterization revealed the presence of genetic markers associated with mammalian infectivity and disease potential. Furthermore, their predicted receptor binding site sequences indicate their potential capacity to attach to human-type receptors. These findings highlight the importance of continued surveillance and molecular investigation to better understand the evolution and zoonotic potential of H9N2 viruses in Kazakhstan.

## 1. Introduction

The rise in humans infected with the avian influenza viruses (AIVs) of the H9N2 subtype underscores the need for a genomic survey of the epidemic potential of their contemporary strains circulating in natural reservoirs and new hosts [1]. A high prevalence of H9N2 in avifauna has been identified as a significant risk for interspecies transmission of these viruses [2,3,4]. Since the late 1990s, the H9N2 subtype of AIVs has been identified in a diverse range of wild bird species and poultry worldwide, and, occasionally, these viruses have expanded their host range by infecting mammalian species [5]. Since 2016, a distinct mammalian lineage of the H9 subtype influenza viruses has emerged in bat populations [1,6].

The initial cases of H9N2 influenza viruses were detected in poultry in China [7]. Subsequently, the virus has disseminated to numerous countries in Asia, the Middle East, and West and North Africa, where extensive poultry farming is practiced, and the virus has become endemic in those poultry populations [8]. The first documented case of human infection with the H9N2 subtype AIV occurred in China in 1998 [9]. Furthermore, twenty-seven additional human cases of influenza A(H9N2) viruses were confirmed in China, Egypt, and Bangladesh in the 5 years between 2013 and 2018, compared to thirteen cases reported in the 14 years prior to 2012 [3,10], indicating an increasing threat to public health. Moreover, recent genetic analyses have revealed that H9N2 AIVs have contributed to the expansion of the genetic and geographic diversity of H5N1 AIV through genetic reassortment [11,12]. Additionally, the emergence of novel H7N9 viruses in 2013, which have infected numerous individuals in China, is attributed to reassortment with six internal genes originating from H9N2 viruses [13]. Furthermore, other subtypes of AIVs, including H5N6, H5N8, and H10N8, may potentially acquire the internal genes of H9N2 viruses through genetic reassortment [14]. The global distribution of H9N2 viruses, with the potential of being capable of infecting mammals and humans, presents a cause for concern regarding their increasing potential to spark a pandemic.

Infection with the H9N2 AIVs in poultry typically manifests as mild clinical signs, including respiratory issues, decreased egg production, and weight loss. Lethal cases are predominantly associated with mixed infections, including bacteria and other viral agents [15]. It has been demonstrated that the H9N2 subtype AIVs may induce transient immunosuppression, potentially exacerbating concurrent or subsequent infections [16]. All H9N2 AIVs are classified as low pathogenic avian influenza viruses (LPAIVs), as they do not cause mortality in an in vivo pathotyping test conducted in specified pathogen-free (SPF) chickens [17].

Genetic studies have demonstrated that the hemagglutinin (HA) of influenza A (H9N2) viruses is closely related to the recently reported H19 subtype of AIV [18], and HA and neuraminidase (NA) genes have split into North American and Eurasian lineages. The North American lineage is exemplified by prototype strain A/Turkey/Wisconsin/1966 (WI/66-like). In contrast, the Eurasian lineage is constituted by a number of distinct sublineages, including BJ/94-like (A/Chicken/Beijing/1/1994-like), G1-like (with A/Quail/Hong Kong/G1/1997 as the prototype), Korea-like (A/Chicken/Korea/38349-p96323/1996), G9-like (A/Chicken/Hong Kong/G9/97), Y439-like (A/Duck/Hong Kong/Y439/1997) and Y280-like (A/Duck/Hong Kong/Y280/97). Most human infections with the H9N2 subtype AIVs are associated with the Y280/G9 lineage. All internal genes of the H9N2 viruses exhibit even greater diversity than the HA and NA genes, with additional sublineages, including the G1-like, F/98-like, (A/Chicken/Shanghai/F/1998) and H5N1-like [19].

Despite large-scale AIV surveillance in Kazakhstan since 2002, the circulation of H9 viruses among wild birds remained undetected until 2014 [20,21]. Since 2020, the H9N2 subtype has been identified as the causative agent of avian influenza outbreaks in poultry farms in the south-eastern region of Kazakhstan.

Three major migratory flyways of wild birds cross through Kazakhstan: Black Sea-Mediterranean, West Asia-East African and Central Asia [21]. The Central Asian and West Asian-East African migration routes are of particular importance, as they encompass flights originating in Siberia and traversing Kazakhstan to India and Africa. The Black Sea-Mediterranean Flyway encompasses most of the Eurasian continent, extending from the Arctic to southern Africa. All three migratory routes traverse North Kazakhstan, and birds may migrate through the Almaty region along the West Asia-East African and Central Asian flyways.

The main objective of this research was to assess the genetic peculiarities of isolates of the H9N2 subtype AIVs isolated from birds in Kazakhstan and their potential for transmission to mammals and humans.

## 2. Materials and Methods

### 2.1. Field Study and Sample Collection

In the context of an AIV monitoring program conducted between 2010 and 2020, a comprehensive sampling effort was undertaken at main migration route transections across Kazakhstan (Figure 1). A total of 5531 cloacal and oropharyngeal swab samples were collected from wild birds and domestic poultry. The swab samples were collected from 3966 wild birds inhabiting aquatic and semi-aquatic ecological complexes. The samples were obtained primarily from hunted waterfowls, live-sampled birds during banding activities, and freshly deposited droppings. AIV-positive fecal samples were additionally subjected to Sanger sequencing of avian COX1 fragments to determine the host species [22,23]. Domestic poultry samples were from 175 commercially breeding broilers and egg layers, 26 domestic waterfowl in the Almaty region, and 59 backyard birds (48 chickens, 6 ducks and 5 geese) in the North Kazakhstan region. The sampling process and specimen handling were conducted in accordance with the methodologies recommended by the World Organization for Animal Health (WOAH) [17].

The samples were obtained using sterile cotton swabs (Nuova APTACA S.R.L., Cod.6100/SG/CS, Canelli, Italy) and transferred into the pre-prepared cryovials (SARSTEDT, Nümbrecht, Germany) containing a virus transport medium containing antibiotics and antimycotics (2000 U/mL penicillin, 2 mg/mL streptomycin, 50 μg/mL gentamicin, and 50 U/mL nystatin), as well as bovine serum albumin at a final concentration of 0.5%. Subsequently, the samples were stored in liquid nitrogen (−196 °C) until testing. The samples were screened for AIVs by a matrix (M) gene-based RT-PCR and/or a generic quantitative real-time RT–PCR (RT-qPCR) targeting the AIV nucleoprotein (NP) gene. Virus isolation was conducted by inoculating the PCR-positive samples into the allantoic cavity of three 10- to 11-day-old embryonated chicken eggs. The eggs were incubated at +37 °C for 48 to 72 h [17].

### 2.2. Sample Preparation and Nucleic Acid Extraction

RNA extraction was carried out on the 140 µL swab sample or harvested allantoic fluids using the QIAamp Viral RNA Mini Kit (Qiagen GmbH, Hilden, Germany), following the manufacturer’s guidelines.

### 2.3. Diagnostic Procedure

To screen the RNA samples for the presence of influenza A virus, a one-step RT-PCR multiplex was performed concurrently, employing the OneTaq^®^ One-Step RT-PCR Kit (New England Biolabs, Ipswich, MA, USA). This multiplex reaction employed 14 pairs of primers [25,26]. To amplify all eight gene segments, 2 or 3 overlapping PCR products were generated to ensure sequence quality from both directions. Following the manufacturer’s protocol, the OneTaq One-Step RT-PCR Kit (New England Biolabs, Ipswich, MA, USA) was employed for the RT-PCR. Subsequently, the PCR products were separated by electrophoresis on 1.5% agarose gels. The RT-PCR products were then excised from the gel and purified using the QIAquick PCR Purification Kit (Qiagen GmbH, Hilden, Germany).

### 2.4. Sequencing

The purified PCR products were sequenced using the BigDye Terminator v3.1 sequencing kit (ABI, Foster City, CA, USA) on an ABI 3500 Genetic Analyzer (Applied Biosystems, Life Technologies, Carlsbad, CA, USA). Sequencing was conducted on both sides of each PCR product. The sequences displaying unique peaks with quality values exceeding 40 were obtained to compare to other H9 avian influenza virus strains from GenBank using the Clustal W algorithm.

RNA from one isolate was subjected to reverse transcription and double-stranded DNA construction utilizing a QIASeq Stranded RNA Library Kit (Qiagen, Hilden, Germany), along with a random hexamer primer at a concentration of 100 pmol. Subsequently, four libraries were generated using the NEB Next Ultra DNA Library Prep Kit for Illumina (New England Biolabs, Ipswich, MA, USA) and sequenced on the MiSeq Illumina platform using the MiSeq Reagent v.3 kit (2 × 300) (Illumina, San Diego, CA, USA). The obtained data were then subjected to trimming and de Novo assembly using Geneious Prime 2022 software (Biomatters, Auckland, New Zealand). Subsequently, the assembled contigs were subjected to a BLASTx search in the local viral reference database, as described in Diamond software v.2.1.6. BLAST. Hits with lengths of more than 200 nucleotides were considered significant at an E-value < 1 × 10^−5^, and the potential viral sequences were subjected to a new BLASTx search in the NCBI non-redundant protein database [27].

### 2.5. Phylogenetic Analysis

The sequences selected from the NCBI database encompassed all H9N2 isolates, representative strains previously documented in the study, and highly pathogenic avian influenza viruses (HPAIV) that exhibited substantial homology with the internal genes of the strains examined in this study. Furthermore, pairwise sequence alignments were conducted using Clustal W.

Phylogenetic trees were constructed using the neighbor-joining approach (MEGA 11.0 software) based on a Tamura–Nei model algorithm [28]. To assess the robustness and consistency of the tree topologies, 1000 bootstrap replicates were employed. The phylogenetic trees were rooted on the sequences of reference strain A/turkey/Wisconsin/1966 (H9N2) as the outgroup to demonstrate their relatedness to the subjected sequences.

## 3. Results

### 3.1. Virological Findings:

Only four of the one-hundred-and-one AIV-positive samples ultimately tested positive for the H9 subtype. Three of the samples (A/Pintail/North Kazakhstan/6368/2014 (Pt/NK/6368/14), A/Mallard/North Kazakhstan/6369/2014 (Ml/NK/6369/14), and A/Whooper Swan/Sorbulak/7994/2019 (Ws/Sb/7994/19)) were isolated from waterfowl in the vicinity of the Sorbulak and Shyoptykol lakes (Table 1). The positive samples from wild birds were obtained from the oropharyngeal and cloacal swabs of live sampled birds in 2014 and 2019. An additional isolate A/Chicken/Almaty/220/2020 (Ck/Ay/220/20) was recovered from the liver tissue of a dead chicken during an outbreak of avian influenza in the Southeastern part of Kazakhstan (Figure 1).

### 3.2. Molecular Diagnostic Findings:

The genome sequences of the H9N2 isolates were compared to other H9N2 sequences in the GenBank and GISAID databases using the nucleotide blast program (BLASTn) to determine the evolutionary relationships among different subtypes of influenza viruses. The epizootic isolate Ck/Ay/220/20 genome exhibited exclusively H9N2-derived segments, indicating that H9N2 poultry strains undergo minimal reassortment with AIV subtypes of waterfowl origin. In contrast, wild waterfowl isolates exhibited diverse gene constellations, representing nine distinct AIV subtype donors.

### 3.3. Phylogenetic Analysis:

The HA and NA genes of H9N2 subtype AIVs isolated in Kazakhstan exhibited high genetic relatedness (99.6% for the HA gene and 99.8% for the NA), except the chicken isolate, Ck/Ay/220/20, which displayed a distinct pattern.

The HA gene of Ck/Ay/220/20 clustered with the H9N2 viruses detected among poultry in Russia, China, and Tajikistan and belongs to the h9.4.2.5/Y280 lineage. The three wild bird H9 sequences clustered with the h9.2/Y439 lineage viruses detected in wild birds and domestic poultry (Figure 2).

The NA gene of Ck/Ay/220/20 clustered with the H9N2 viruses detected in poultry in China and belongs to the Y280 lineage, and the three wild bird H9N2 sequences clustered with the Y439 lineage viruses detected in wild birds and ducks found mainly in Asia (Figure 3).

The phylogenetic analyses of internal genes (Appendix A) indicated that wild bird H9N2 viruses displayed high genetic relatedness (97–99%), and all genes belonged to the Y439 lineage except PB2, which belonged to the G1 lineage. The four internal genes of Ck/Ay/220/20, including PB1, PA, NP, and NS, clustered within the SF98 lineage and were closely related to the poultry isolates from Tajikistan and Russia. However, the M and PB2 genes were grouped into the G1 lineage and exhibited close relatedness to poultry isolates from China in 2018–2019.

### 3.4. Molecular Characterization

The four studied H9N2 subtype AIVs possessed various molecular markers associated with the adaptation of influenza A viruses in mammals. These markers included nine amino acid (aa) substitutions in PB2, four in PB1, seven in PA, two in NP, five in M1, and six in NS1 (Appendix A).

#### 3.4.1. Hemagglutinin (HA)

A sequence alignment analysis of the HA protein of the studied viruses revealed the presence of specific genetic markers in the HA receptor-binding site (RBS), including S127N, H183N, Q234L, and 236G (Table 2). The S127N mutation was observed in wild bird H9N2 isolates, which have been demonstrated to be associated with a potentially higher level of pathogenicity, particularly for specific pathogen-free (SPF) chickens [29,30]. The aa substitution Q234L in Ck/Ay/220/20 has been demonstrated to promote attachment to human receptors [31]. All four isolates exhibited the 236G substitution, associated with human virus-like cell tropism [32]. Moreover, three additional molecular markers with mammalian-like motifs were identified in the Ck/Ay/220/20 isolate: S158N, K183S, and N198T (Appendix A) [30,33].

The HA protein’s potential N-linked glycosylation sites of the studied H9N2 viruses exhibited eight *N*-linked glycosylation sites with the N-X-T/S motif (where X can represent any aa except proline) [34]. The potential N-linked glycosylation sites were identified at the following positions: 29, 82, 141, 218, 298, 305, 313, 492, and 551 (Figure 4 and Table 2). Despite each virus having eight N-linked glycosylation sites, only six occurred in all four viruses, and one site differs between the wild bird and chicken H9N2 viruses.

Three unreported substitutions (V12I, T38M, and N153S) were identified in wild bird H9N2 studied isolates, and other substitutions, L51I, M187I, A373V, and I422V, were exclusively identified in Ck/Ay/220/20.

#### 3.4.2. Neuraminidase (NA)

Phylogenetic analysis of the NA gene revealed a similar evolutionary pattern to that observed for the HA genes. Three wild bird viruses in the study clustered within the Y439-like lineages. However, Ck/Ay/220/20 was clustered within the Y280-like lineage. The NA protein of the investigated H9N2 AIVs exhibited an aa deletion in the stalk region, which has been demonstrated to enhance virus adaptation from wild birds to poultry [35]. Furthermore, Ck/Ay/220/20 and all viruses within the Y280-like lineage exhibited a three aa deletion motif at positions 63–65, as previously documented [36]. This deletion pattern is consistent with the sequences of poultry H9N2 viruses prevalent in mainland China (Appendix A). Furthermore, the A30T substitution in Ck/Ay/220/20 (Appendix A) has been previously documented in ferrets infected with H9N2 [37]. The V165I substitution in the NA gene was observed exclusively in wild bird H9N2 isolates. In addition, the following substitutions were identified solely in this study: M210I in Pt/NK/6368/14 and K241E, T385I, and G414S in Ck/Ay/220/20.

#### 3.4.3. Polymerase Basic 2 (PB2)

The PB2 protein of most H9N2 isolates exhibited substitutions such as L89V, G309D, K318R, T339K, R477G, I495V, I504V, E627V, and A676T, which have been identified as markers of mammalian adaptation [37,38,39,40,41]. Notably, the A588V substitution, which has recently been recognized as a marker of adaptation of AIVs to mammalian species [36], was observed in the Ml/NK/6369/14 and Ck/Ay/220/20 isolates. Moreover, all the Y439-like viruses in this study exhibited the K389R mutation in the PB2 protein. Additionally, unreported substitutions, including E65G, Q138L, T371A, A401G, and A468T, were exclusively identified in Ck/Ay/220/20.

#### 3.4.4. Polymerase Basic 1 (PB1)

The key PB1 residues responsible for adaptation of avian influenza viruses to mammalian hosts [6,42], such as D3V, L13P, and D622G (Appendix A), were found in four H9N2 studied isolates. The aa substitution I368V was only identified in Ck/Ay/220/20. Additionally, unreported substitutions, including G352R, I355Y, A461R, and G516W, were detected in Pt/NK/6368/14 and I102D in Ck/Ay/220/20.

#### 3.4.5. Polymerase Acidic (PA)

In the case of Ck/Ay/220/20, two additional molecular markers with mammalian-like motifs in the PA protein (at positions V63I and K356R) [6] were identified. The key PA residues responsible for the adaptation of avian influenza viruses to mammalian hosts, leading to increased polymerase activation, such as K26E, V160D, and N383D (Appendix A), were found in all four of the studied H9N2 isolates [6,43,44]. Molecular markers (S37A and N409S) affecting increased polymerase activity in mammalian cell lines were found in the PA protein H9N2 isolates from wild birds [44]. Unreported substitutions, including S395C substitution only in Ml/NK/6369/14, and E56G, R196S, S277F, N350K, and P534T substitutions, were identified in Ck/Ay/220/20 PA protein.

#### 3.4.6. Nucleoprotein (NP)

The NP protein of four H9N2 isolates exhibited two substitutions, K398Q and 136L, which have been identified as markers associated with adaptation to mammalians [35,41] (Appendix A). Unreported substitutions, A323T and D468H (Appendix A), were identified in Pt/NK/6368/14 isolate, while R121G, N124K, H135P, and C279S substitutions were identified exclusively in Ml/NK/6369/14 isolate.

#### 3.4.7. Matrix (M)

The M1 protein of all studied H9N2 isolates exhibited markers of mammalian infection—N30D, I43M, and T215A—which increased virulence in mice [6] (Appendix A). Furthermore, the Ck/Ay/220/20 isolate exhibited specific mammalian transmission (V15I in M1; I28V and I55F in M2) and elevated virulence (T37A, R95K, S224N, and K242N in M1, and D21G in M2) markers [35]. Moreover, the last five residues are characteristic of G1-like H9N2 viruses’ M genes [45].

#### 3.4.8. Nonstructural Protein (NS1)

The P42S, K66E, C138F, and V149A mutations in the NS1 protein [6,46] were identified in all strains. Furthermore, three molecular markers K55E, L103F, and I106M with mammalian-like motifs were identified in the NS1 protein of H9 isolates from wild birds, and E227K was identified in Ck/Ay/220/20 (Appendix A). Additionally, the previously undocumented S135R and K217E substitutions in the NS1 gene were identified in the Pt/NK/6368/14 and Ml/NK/6369/14 isolates.

## 4. Discussion

AIVs, particularly HPAIVs, are receiving increasing attention due to their zoonotic nature and transmission mode. Transmission of H9 subtype AIVs to humans has been reported previously [9]. The critical importance of H9N2 viruses at the animal–human interface may be due to their wide host range, adaptation to both poultry and mammals, and extensive gene reassortment. The seasonal migration of birds plays a significant role in the long-distance transmission of the AIVs [47]. Indeed, the three longest-distance migratory flyways—the Black Sea-Mediterranean, West Asia-East Africa, and Central Asia—cross Kazakhstan [48]. To better understand the global evolutionary dynamics of H9N2 subtype AIVs, Jiang et al. [49] categorized them into four central stem evolutionary clades, designated as h9.1 through h9.4.

In this study, three H9N2 viruses, designated Pt/NK/6368/14, Ml/NK/6369/14 and Ws/Sb/7994/19, were isolated from wild waterfowl and classified into the Y439-like lineage (clade h9.2). Furthermore, Ck/Ay/220/20 was the first H9N2 from the Y280-like lineages (clade h9.4.2.5) to be identified in Kazakhstan. The phylogenetic analysis of the HA and NA genes indicated that Ck/Ay/220/20 belonged to the Y280-like lineage of viruses isolated in the bordering countries of Kazakhstan, particularly China, Russia, and Tajikistan [50]. However, its M gene originated from G1-like viruses, while its PA, NP, and NS genes were from SF98-like viruses. Notably, the Y439-like lineage viruses in the present study had only acquired the PB2 gene from the G1-like lineage viruses. This suggests that a specific gene exchange event or reassortment involving the PB2 gene may have occurred.

The aa pattern at the cleavage site of the HA protein is an essential indicator of pathogenicity [51]. Although the cleavage site motifs of the four H9N2 isolates exhibited two distinct patterns, both were of low pathogenicity cleavage motifs. The HA protein RBS motif is crucial in determining the host range of influenza viruses, with specific aa in RBS determining its specificity. AIVs typically recognise sialyl α2,3-galactose receptors, whereas human influenza viruses recognise sialyl-α2,6-galactose receptors [52]. A recent study revealed that the H9N2 viruses isolated from chickens exhibited leucine (Leu) and glycine (Gly) at positions 234 and 236 in the RBS [35], and the H9N2 viruses isolated from wild birds in this study also exhibited asparagine (Asn), Asn, and Gly at positions 127, 183, and 236 in the RBS. Notably, three H9N2 isolates from wild waterfowl possessed a Gly at position 236, which is associated with human cell tropism characteristic of H9N2 AIVs [35]. Furthermore, the poultry isolate, Ck/Ay/220/20, contained Leu at position 234, a known contributor to human-type receptor binding in H9N2 viruses [53]. Studies have demonstrated that the human-specific RBS, particularly the combination of 183N and 234L, may contribute to the accumulation of adaptive mutations in the HA protein of the Kazakhstan’s H9N2 isolate(s) and subsequent healthcare risk potential. The studies demonstrated that viruses with the 127N residue site have an enhanced propensity to bind to human-like receptors [53].

Importantly, the identified patterns of potential N-linked glycosylation sites in the current study were consistent with those observed in most previously reported H9N2 viruses [36,54]. A nonconserved N-linked glycosylation site at position 145 in the globular domain of the HA protein was absent in all studied viruses. Additionally, three wild bird H9N2 AIVs lost an extra N-linked glycosylation site at position 313 due to an S315P substitution; however, this has been apparently compensated via an additional potential N-linked glycosylation site at position 218, a feature also detected in human H9N2 influenza viruses A/Hong Kong/1074/1999 and A/Hong Kong/3239/2008 [55]. These findings elucidate the molecular characteristics of the HA protein and its potential significance for host specificity and adaptation.

The human H9N2 isolates in Vietnam in 2018 exhibited deletions in the stalk region of the NA protein at positions 62–64, which have been associated with mammalian adaptation [56]. The deletion was observed in the Ck/Ay/220/20 isolate.

The viral polymerase complex, composed of PB2, PB1, PA, and NP proteins, plays a pivotal role in the potential adaptation of AIVs to mammalian hosts. A number of aa substitutions in these proteins have been identified that contribute to increased polymerase activity, replicative ability, and virulence in mammalian models [57,58,59]. It is important to highlight that the recently identified E627V substitution in PB2 as a mammalian-adaptation mutation in H5N1 clade 2.3.4 viruses [39] was found in the A/220/20 and A/6369/14 isolates. A similar substitution has also been identified in human H9N2 influenza viruses A/Beijing/1/2016 and A/Beijing/1/2017 [52]. All the Y439-like isolates in the present study exhibited the K389R mutation in the PB2 protein, which has been associated with increased polymerase activity and replication in mammalian cell lines [6,36]. These findings provide valuable insights into the potential for adaptation and enhanced virulence of these H9N2 avian influenza viruses in mammalian hosts. 

The D3V and D622G substitutions in PB1 have been demonstrated to enhance polymerase activity in mammalian cell lines [6]. Furthermore, the L13P substitution was associated with mammalian adaptation, and the virus is able to infect mice and is associated with enhanced virulence in mammalian species [60]. The I368V substitution was identified exclusively in Ck/Ay/220/20, which has been demonstrated to be linked to the adaptation of AIVs to a novel host species [54]. Furthermore, the L13P substitution was associated with mammalian adaptation and a virus with this substitution is able to infect mice.

The substitutions K26E, V160D, and N383D were identified in the PA protein of all tested H9 viruses. These substitutions have been demonstrated to be associated with increased polymerase activity and replication in mammalian cell lines [6,43,44]. The N383D substitution in the PA protein is suggestive of enhanced virulence in ducks [61]. Furthermore, markers linked to mammalian hosts, such as S37A and N409S, were identified in PA protein of Pt/NK/6368/14, Ml/NK/6369/14, and Ws/Ay/7994/19 isolates [44].

Previous studies demonstrated the association of N30D, I43M, and T215A substitution in the M1 protein [6] and the P42S, K66E, C138F, and V149A substitutions in the NS1 protein [6,46] with increased virulence of the virus in mice. Notably, the aforementioned substitutions were detected in four H9N2 isolates from the current study. Furthermore, the NS1 protein of isolates from wild birds exhibited two additional molecular markers with mammalian-like motifs, L103F and I106M [62]. In the NS1 protein of the studied viruses, the identified markers of K55E, K66E, and C138F may cause higher virus replication and lower level of interferon response in mammalian cells [40].

A comprehensive genetic analysis of four H9N2 subtype AIVs isolated and characterized within the current study revealed that the genome of all tested H9N2 viruses has exhibited numerous genetic markers associated with potential adaptation of these viruses to mammalian hosts, including humans, and higher pathogenicity in mammalian species. The current co-circulation of the H9N2 AIVs and H5Nx subtypes (particularly clade 2.3.4.4b) HPAIVs in birds may facilitate the emergence of novel viruses that are capable of infecting mammals and/or exhibiting a higher level of virulence and pathogenicity for avian and mammalian species.

The current state of knowledge regarding the AIV H9N2 viruses circulating among wild waterfowl and domestic poultry in Kazakhstan is limited. Furthermore, the phylogenetic and epidemiological links between natural and epizootic isolates remain understudied. In the current study, AIV H9N2 isolates from wild waterfowl in Kazakhstan demonstrated a resemblance to a cluster of mallard viruses from Russia and viral variants from dabbling ducks [63] of the Iranian coast of the Caspian Sea. This emphasizes the importance of the Caspian region as a prominent spot where many bird migration flyways intersect, and different viral lineages and genotypes may be intermingling [1].

Further research is essential to characterize these viruses and assess their transmission potential through reassortment with H5Nx viruses to understand better the zoonotic risks they pose. Moreover, the close genetic relatedness of Kazakhstan strains to those from neighbouring countries suggests cross-border transmission, emphasizing the necessity for regional surveillance and coordinated efforts to control the spread of the virus.

## Figures and Tables

**Figure 1 viruses-17-00685-f001:**
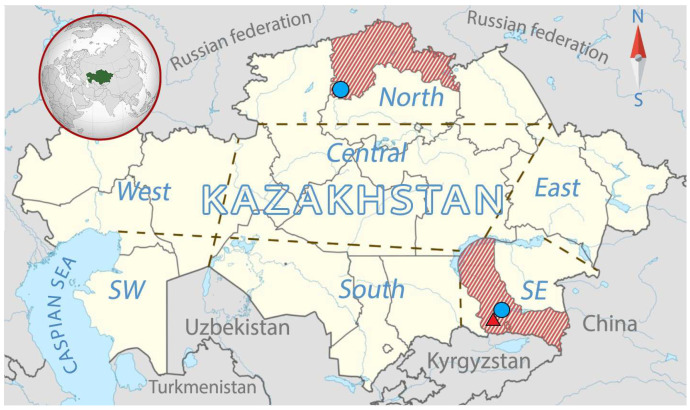
Map of sampling sites in this study. The two regions where samples were collected are indicated by a red dash (the map was modified from [24]). The blue circles indicate the locations of lakes where H9N2 strains were isolated from waterfowl in the North Kazakhstan and Almaty regions. The red triangle indicates the location of the poultry farm in the Almaty region.

**Figure 2 viruses-17-00685-f002:**
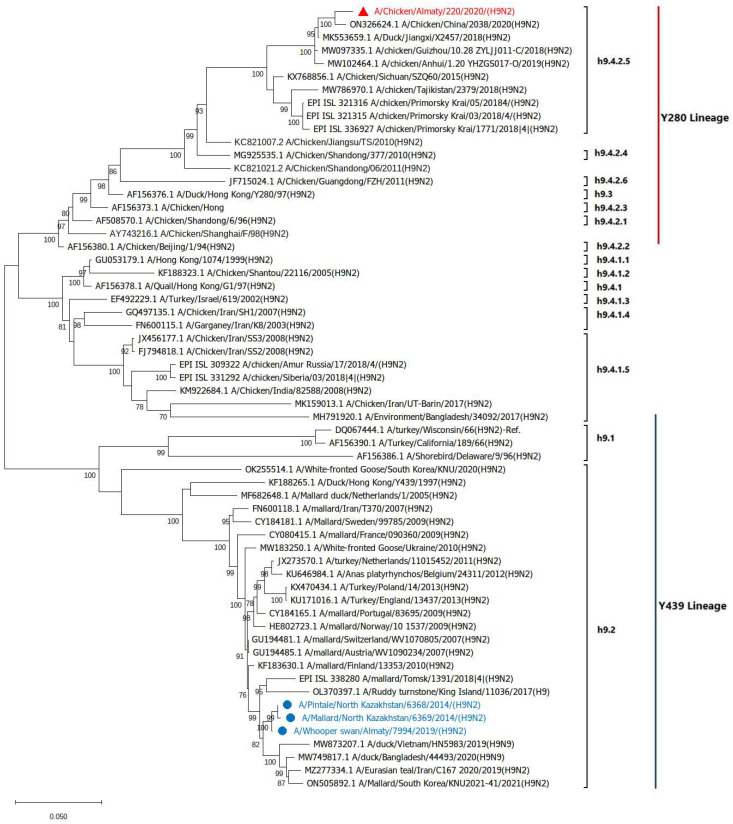
Phylogenetic tree of the full-length HA sequence of the H9N2 isolates from Kazakhstan, including A/Chicken/Almaty/220/2020 (marked with red triangle) and wild bird’s isolates (marked with blue dots) and other related sequences from wild and domestic birds (GenBank and GISAID databases). Numbers at nodes indicate maximum likelihood bootstrap values of 1000 replicates under the specified model. Only the bootstrap values above 70 are shown. The tree is rooted to A/turkey/Wisconsin/1966. Bar, 0.05 nt substitutions per site.

**Figure 3 viruses-17-00685-f003:**
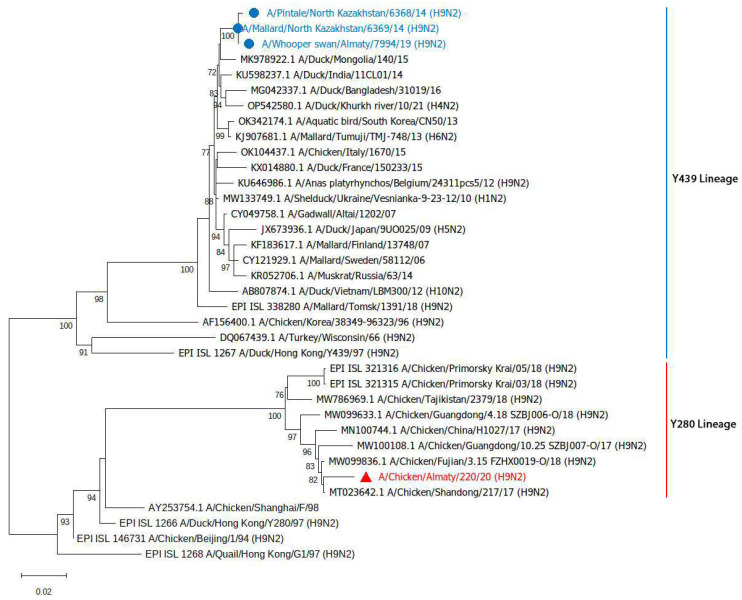
Phylogenetic tree of the full-length NA sequence of the H9N2 isolates from Kazakhstan, including A/Chicken/Almaty/220/2020 (marked with red triangle) and wild birds isolates (marked with blue dots) and other related sequences from wild and domestic birds (GenBank and GISAID databases). Numbers at nodes indicate maximum likelihood bootstrap values of 1000 replicates under the specified model. Only the bootstrap values above 70 are shown. The tree is rooted to A/turkey/Wisconsin/1966. Bar, 0.02 nt substitutions per site.

**Figure 4 viruses-17-00685-f004:**
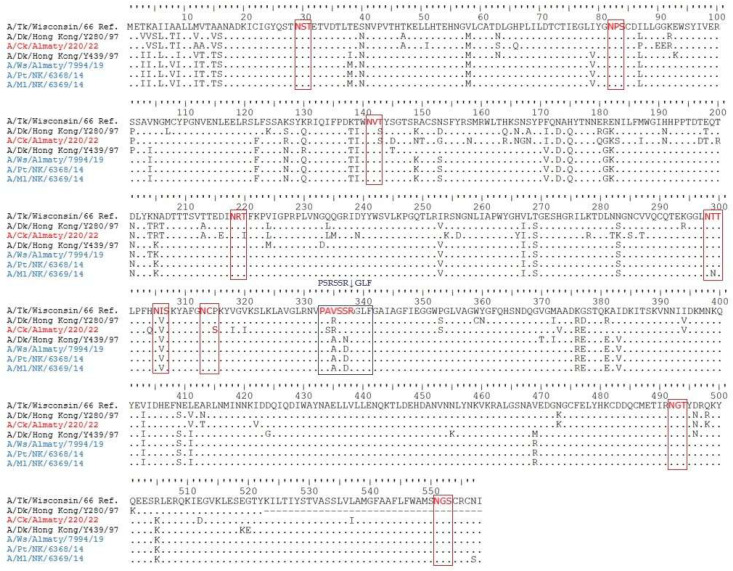
Alignment of HA aa sequences of four of Kazakhstan’s IAVs and their consensus sequence in comparison with the reference viruses’ IAVs and their consensus sequence. Cleavage and glycosylation sites are indicated in blue and red boxes, respectively.

**Table 1 viruses-17-00685-t001:** The spatial–temporal data of Kazakhstan’s Avian Influenza H9N2 viruses involved in this study.

AIV H9N2 Isolate and Sample Collection Date	Abbreviation	Accession Number of Sequence in GenBank	Collection Site, Geographic Location
A/Pintail/North Kazakhstan/6368/2014 (October 2014)	Pt/NK/6368/14	PV390943-PV390951	North Kazakhstan, Shyoptykol lake 52°24′01.9″ N, 66°50′05.5″ E
A/Mallard/North Kazakhstan/6369/2014 (October 2014)	Ml/NK/6369/14	PV390827-PV390835	North Kazakhstan, Shyoptykol lake 52°24′01.9″ N, 66°50′05.5″ E
A/Whooper Swan/Sorbulak/7994/2019 (October 2019)	Ws/Sb/7994/19	PV390876-PV390884	South-East Kazakhstan, Lake Sorbulak 43°38′48.4″ N, 76°32′20.5″ E
A/Chicken/Almaty/220/2020 (August 2020)	Ck/Ay/220/20	PV390766-PV390774	South-East Kazakhstan, Almaty suburb 43°21′15.3″ N, 76°51′48.6″ E

**Table 2 viruses-17-00685-t002:** Molecular characteristics of the hemagglutinin amino acid sequences in H9N2 AIV in this study.

Strains	Receptor-Binding Sites				Cleavage Peptides	N-Linked Potential Glycosylation Sites (H9 Numbering)								
	S* 127N	H 183N	Q 234L	236G		29–31	82–84	141–143	218–220	298–300	305–307	313–315	492–494	551–553
Pt/NK/6368/14	+	+	Q	+	PAASDR↓GLF	NST	NPS	NVT	NRT	NTT	NVS	—	NGT	NGS
Ml/NK/6369/14	+	+	Q	+	PAASDR↓GLF	NST	NPS	NVT	NRT	NNT	NVS	—	NGT	NGS
Ws/Ay/7994/19	+	+	Q	+	PAASDR↓GLF	NST	NPS	NVT	NRT	NTT	NVS	—	NGT	NGS
Ck/Ay/220/20	R	S	+	+	PSRSSR↓GLF	NST	NPS	NVS	—	NTT	NVS	NCS	NGT	NGS

* A, alanine; C, cysteine; G, glycine; F, phenylalanine; I, isoleucine; L, leucine; N, asparagine; P, proline; Q, glutamine; R, arginine; S, serine; T, threonine; V, valine; +, Presence of specific aa substitutions; —, absence of glycosylation site.

## Data Availability

Sequence data were submitted to GenBank. The accession numbers are provided in the manuscript.

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
