# Peer review of "Genetic Characterization of Kazakhstan Isolates: Avian Influenza H9N2 Viruses Demonstrate Their Potential to Infect Mammals"

_viruses, 2025, doi:10.3390/v17050685_

Round 1

Reviewer 1 Report

Comments and Suggestions for Authors

This study delineates the molecular genetic characteristics of H9N2 avian influenza A viruses isolated across various regions of Kazakhstan from 2014 to 2020. Molecular analyses revealed that the examined H9N2 strains harbor genetic markers indicative of their potential to infect mammals and induce pathogenesis. Furthermore, predicted receptor-binding site (RBS) configurations suggest these viruses possess human-specific receptor-binding capabilities. These findings underscore the critical need to investigate the origin and evolutionary trajectory of H9N2 viruses within Kazakhstani poultry populations. Collectively, the methodological rigor and analytical depth of this work meet acceptable scientific standards, warranting consideration for publication.

Author Response

Comment: This study delineates the molecular genetic characteristics of H9N2 avian influenza A viruses isolated across various regions of Kazakhstan from 2014 to 2020. Molecular analyses revealed that the examined H9N2 strains harbor genetic markers indicative of their potential to infect mammals and induce pathogenesis. Furthermore, predicted receptor-binding site (RBS) configurations suggest these viruses possess human-specific receptor-binding capabilities. These findings underscore the critical need to investigate the origin and evolutionary trajectory of H9N2 viruses within Kazakhstani poultry populations. Collectively, the methodological rigor and analytical depth of this work meet acceptable scientific standards, warranting consideration for publication.

Response: We want to thank Reviewer 1 for the warmly positive review of our submitted manuscript, recommending it for publication.

Reviewer 2 Report

Comments and Suggestions for Authors

The manuscript describes the genetic characterisation of avian H9N2 subtype influenza A viruses in Kazakhstan.

I suggest restructuring the abstract and providing the context / importance of monitoring for avian IAV in the region due to the flyways first as the background to the study, correcting some English, avoiding the use of an unexplained abbreviation (RBS) and making it clear that all eight gene segments (not just HA or HA & NA were sequenced):

“Low pathogenic H9N2 avian influenza viruses have become widespread in wild birds and poultry worldwide, raising concerns about their potential to cause pandemics or contribute to higher virulence and infectivity of H5Nx viruses via genetic reassortment. Therefore, influenza monitoring studies, including H9N2 viruses, are crucial for understanding, evaluating, and mitigating the risks associated with avian infections and have broader implications for global public health. Although H9N2 viruses are not enzootic in Kazakhstan, which is crossed by three major wild bird flyways, they have been repeatedly detected in wild waterfowls and domestic poultry.

In this study, all eight gene segments of influenza A/H9N2 viruses isolated in different regions of Kazakhstan from 2014 to 2020 were sequenced. The molecular analysis has revealed that the studied H9N2 avian influenza viruses have genetic markers demonstrating their potential to infect and cause disease in mammals. Additionally, their predicted receptor binding site sequences indicate their capacity to attach to human-specific receptors. These findings underscore the urgent need to investigate the origin and evolution of H9N2 viruses in poultry in Kazakhstan.”

I am not sure that the correct references are always cited. For example on line 79 it is stated that the circulation of H9 viruses among wild birds (in Kazakhstan?) remained undetected until 2014 but the first reference cited is about H10N8 in feral dogs in China.

In the methods it states that phylogenetic trees were midpoint rooted, which is true for figure 2, but figure 3 is rooted on A/turkey/Wisconsin/1966 – either make both midpoint rooted or explain why one was rooted to an outgroup.

Line 235 – delete ‘none of the isolates contained the A190T substitution as it is not clear what the significance of this is – this should be in the discussion if it is described that this is usually associated with mammalian adaptation?

Minor typographical / English language corrections:

Line 32: The rise in human infections with the avian influenza viruses (AIVs) H9N2 subtype - with avian influenza A viruses (AIVs) of the H9N2 subtype…

Line 37: and in occasional conditions – and occasionally

Line 41: By the early 1990s, the initial – The initial…

Line 47: and Bangladesh between 2013 and 2018, rather than 13 cases reported prior to 2012 - and Bangladesh in the 5 years between 2013 and 2018, compared to13 cases reported in the 14 years prior to 2012

Line 55: - should this be potentially capable, rather than partly capable

Line 58: manifests as mild clinical symptoms – clinical signs (veterinary species do not have symptoms)

Line 64: conducted in specified (not ‘on’)

Line 67 – closely related to the recently reported H19 subtype of AIV

Line 75 – All internal genes of the H9N2 viruses

Line 83 – Central Asia (not Asian)

Line 96 – main migration route transections

Line 98 – from wild birds and domestic poultry (not wild chickens!). Swab samples were…

Line 102- Domestic poultry samples were from 175 commercial breeding…

Line 110 – the map was modified from [24]

Line 113 – need to give the actual company and number for the sterile cotton swabs!

Line 119 – That 101 samples were found to be positive for presence of AIV and four were isolates were H9N2 should be in the results.

Line 123 – embryonated hens’ eggs (chickens are immature and cannot lay eggs)

Line 124 –‘This study…four isolates’ – delete = not method.

Line 134 – 2 or 3

Line 133-146 = results?

Line 147 – RNA from one isolate (not from original sample?) was….

Section 2.5. start with Sequences selected from..Clustal W alignment then  then phylogenetic trees were constructed…

Lines 172 onwards – suggest it would be clearer to describe the wild bird sequences then the domestic poultry positive samples:

Three of the samples (A/Pintail/North Kazakhstan/6368/2014 (Pt/NK/6368/14),

A/Mallard/North Kazakhstan/6369/2014 (Ml/NK/6369/14), and A/Whooper Swan/Sorbulak/7994/2019 (Ws/Sb/7994/19)) were isolated from waterfowl in the vicinity of the Sorbulak and Shyoptykol lakes (Table 1). The positive samples from wild birds were obtained from the oropharyngeal and cloacal swabs of live sampled birds in 2014 and 2019. An additional isolate A/Chicken/Almaty/220/2020 (Ck/Ay/220/20) was recovered from the liver tissue of a dead chicken during an outbreak of avian influenza in the Southeastern part of Kazakhstan (Figure 1).

Table 1 – Spatiotemporal data of H9N2 subtype avian influenza A viruses isolated in this study.

Suggest giving only month and year for all four samples.

Line 184 – The genome sequences of the H9N2 isolates were compared to other H9N2 sequences…

Line 194 0 chicken isolate (singular)

Line 197 – wild bird H9 sequences

Line 201 (figure 2 legend – same for figure 3) – the chicken isolate is marked with a red triangle not a dot.

I found the text giving the lineages (Y280 and Y439 too small to read easily (similarly for the other labels next to clades…if possible make the text larger.

Lie 204 – above 70%

Line 207 – wild bird H9N2

Line 218 – that the wild bird H9N2 viruses

Line 226 – influenza A viruses (or AIV)

Line 238 – Additionally…additional…

Table 2 – why is Pt/NK/6368/14 in bold typeface? What does ‘+ mean in the receptor-binding site locations? Presumably N, N, L, G is a consensus sequence that is compared to? Or is it the sequence in Pt/NK/6368/14, in which case give it in that line and indicate that this is the same in each of the other two wild bird viruses. Explain what the dash means (absence of glycosylation site)

Line 245 – The potential N-linked glycosylation sites (PGS) in the HA proteins of the …why use PGS and not NGS as an abbreviation for N-linked glycosylation site? If this stands for potential glycosylation site, need to give that in full at first mention. Also give aa in full at first mention. Finally, could be made clearer that although there are eight PGS in each of the viruses, there are only six that occur in all four viruses (i.e. one site differs between the wild bird and chicken viruses).

Line 253 – Previously unreported substitutions? Are these including (i.e there are others) or are there only three that are previously unreported? In the wild bird H9N2 isolates

Line 257 – Phylogenetic analysis…

Line 271 – ‘such as’ – are there others? Why only mentioning these?

Line 280 -The key PB1 residues…

Line 290 – found in all four of the studied…

Line 291 mammalian cell lines (S37A and N409S)…

Line 292 – included S395C only in … substitutions in the Ck…

Line 303 – exhibited markers of mammalian…

Line 310 – the sentence ’Furthermore, additional molecular markers … ‘ is unclear – is there one in the chicken isolate (E227K) and three in the wild bird isolates?

Line 316 – need to give HPAIV in full at first mention.

Line 329 – age  … Phylogenetic analysis…in countries bordering with Kazakhstan

Line 338 – both were low pathogenicity cleavage motifs

Line 351 – Studies have demonstrated

Line 353 – in the current study

Line 354 – this is a fragment of a sentence ‘Noncon- 354

served PGS at position 145 in the globular domain of the HA protein The’

Line 355 – how do you know that the additional PGS at position 218 compensates for the loss of the PGS at 314 – perhaps say ‘apparently compensated’? Does the human H9N2 virus reported in 55 lose / gain the same glycosyation sites?

Line 379: The D3V and D622G substitutions have been demonstrated to enhance polymerase activity in mammalian cell lines of PB1 protein [6].  – The D3V and D622G substitutions in PB1 have been demonstrated to enhance polymerase activity in mammalian cell lines. Furthermore, the L13P substitution was associated with mammalian adaptation and virus with this substitution is able to infect mice …

Line 391 – Previous…substitutions…

Lines 399 – 404 – please delete – as you have not calculated nt substitution rate, this is not directly relevant to your study.

Line 416 – In the current study, AIV H9N2 isolates from wild waterfowl in Kazakhstan demonstrated … mallard

Lines 421 – 429 – this adds nothing and should be deleted / rewritten…this study does not demonstrate the capacity of H9N2 viruses to adapt to mammalian hosts (no infection studies of mammalian hosts were conducted / the viruses were isolated from birds).

Comments on the Quality of English Language

The quality of English language is good (especially considering likely to be second language for all authors) but could be improved in places (as per comments made).

Author Response

RESPONSES TO REVIEWER 2’S COMMENTS

We would like to express our gratitude to reviewer 2 for constructive comments and suggestions, which have been instrumental in enhancing the quality of our work.

General comment regarding Abstract. We modified the abstract based on reviewer’s suggestion

Comment 1. I am not sure that the correct references are always cited. For example, on line 79 it is stated that the circulation of H9 viruses among wild birds (in Kazakhstan?) remained undetected until 2014 but the first reference cited is about H10N8 in feral dogs in China. [2, 20].

Response 1. Thank you for this significant remark. We replaced the improper reference with a correct one.

Comment 2. In the methods it states that phylogenetic trees were midpoint rooted, which is true for figure 2, but figure 3 is rooted on A/turkey/Wisconsin/1966 – either make both midpoint rooted or explain why one was rooted to an outgroup.

Response 2. We appreciate this valuable comment. According to the comment, the statement on methods was revised and rewritten as follows: the phylogenetic trees were rooted on the sequences of reference strain A/turkey/Wisconsin/1966 (H9N2) as outgroup to demonstrate their relatedness to the subjected sequences.

Line 235. “Additionally, they possessed K183N substitution, while none of the isolates contained the A190T substitution” was deleted.

Minor typographical / English language corrections:

Line 32: “The rise in human infections with the avian influenza viruses (AIVs) H9N2 subtype” changed to ‘with avian influenza A viruses (AIVs) of the H9N2 subtype…’

Line 37: ‘and in occasional conditions’ changed to “and occasionally”

Line 41: “By the early 1990s” was deleted.

Line 47: Suggested correction was accepted.

Line 55: The suggestion was accepted.

Line 58: ‘clinical symptoms’ changed to ‘clinical signs’.

Line 64: corrected.

Line 67: The suggestion is accepted.

Line 75: ‘the H9N2’ inserted.

Line 83: corrected to Asia.

Line 96: corrected.

Line 98: The corresponding correction was made.

Line 102: The suggestion was accepted.

Line 110: The suggestion was accepted.

Line 113: The actual company and number for the sterile cotton swabs are provided.

Line 119: The sentences “Consequently, 101 samples were found to be positive for the presence of AIV. Furthermore, genome-based subtyping enabled the identification of four isolates as H9N2 subtype AIV.” were removed from the methods section.

Line 123: In this particular instance, the term "chicken" is used as the species name.

Line 124: The sentence was deleted.

Line 134: inserted ‘or’ between 2 and 3.

Line 133-146: This text fragment describes RT-PCR performing and sequencing method. The fragment “The majority of the sequences displayed unique peaks with quality values exceeding 40. Subsequently, these obtained sequences were compared to other H9 avian influenza virus strains from GenBank using the Clustal W algorithm” which was rewritten in a methodological manner like “The sequences displayed unique peaks with quality values exceeding 40, which were obtained to compare to other H9 avian influenza virus strains from GenBank using the Clustal W algorithm”.

Line 147: “The A/Whooper Swan/Sorbulak/7994 virus RNA” changed to “RNA from one isolate was…”.

Section 2.5. was corrected according to the suggestions.

Lines 172 onwards - rewritten according to your suggestion.

Table 1: Provided only the month and year for all four samples.

Line 184: Correction was made as suggested.

Line 194: corrected to singular.

Line 197. Corrected to singular.

Line 201. Red dots changed to red triangles in Figures 2 and 3.

Request: I found the text giving the lineages (Y280 and Y439 too small to read easily (similarly for the other labels next to clades…if possible make the text larger.

Response: The requested text is enlarged for an easy-to-read size.

Line 204: % was inserted.

Lines 207, 218: corrected to singular form.

Line 226: changed to “influenza A viruses”.

Line 238 – ‘Additionally’ was changed to ‘Moreover’…

Table 2 -Pt/NK/6368/14 turned standard typeface. N, N, L, G is a genetic marker in the amino acid positions 127, 183, 234, 236. The description of these genetic markers and citing Table 2 were made in lines 232-234. The dash means the absence of a glycosylation site; the corresponding correction was in the footnote of Table 2.

Line 245: We decided not to abbreviate “potential N-linked glycosylation sites.” It is mentioned only six times in the whole text of the manuscript. The Complete spelling of aa was given at the first mention in line 229. Your suggestion to summarise this part was made as follows: Despite each virus having eight N-linked glycosylation sites, only six occur in all four viruses, and one site differs between the wild bird and chicken H9N2 viruses (lines 250-251).

Line 253 – The question was elucidated as follows: “Three unreported substitutions (V12I, T38M, and N153S) were identified in wild birds H9N2 studied isolates, and other substitutions, L51I, M187I, A373V, and I422V, were exclusively identified in Ck/Ay/220/20.”

Line 257 – “The” was deleted.

Line 271 – The sentence was rewritten for clarity: “Besides, the following substitutions were identified solely in this study: M210I in Pt/NK/6368/14 and K241E, T385I and G414S in Ck/Ay/220/20.”

Line 280 – “The key PB1 residues” inserted (Line 283).

Line 290 – Corrected according to the suggestion.

Line 291: (S37A and N409S) inserted after “Molecular markers” (Line 294).

Line 303 – Suggestion is accepted as “exhibited markers of mammalian infection”.

Line 310 – The sentence was paraphrased for clarity. “Furthermore, three molecular markers K55E, L103F and I106M with mammalian-like motifs were identified in the NS1 protein of H9 isolates from wild birds, E227K in the Ck/Ay/220/20”.

Line 316 – HPAIV is defined in line 162.

Line 329 – Corrected according to the suggestion.

Line 338 – Corrected according to the suggestion.

Line 351 – Corrected according to the suggestion.

Line 353 – “the” was inserted.

Line 354 – this is a fragment of a sentence, ‘Nonconserved’ at line 354.

Line 355—The used version was ‘apparently compensated’. Only one human H9N2 virus reported in 55 had lost/gained the same glycosylation sites.

Line 379: Suggestion was accepted.

Line 391 – “The” was inserted. Started from “Previous…”

Lines 399 – 404 – We agree with your comment, and the corresponding part was removed from the text.

Line 416 – The suggestion for rephrasing was accepted. The Bird name ‘Mallard’ still capitalized according to the recommendations of the American Society of Ornithology.

Lines 421 – 429 – The text in the indicated lines was partially deleted and rewritten with a focus on the necessity to characterise these viruses and assess their transmission potential through reassortment with H5Nx viruses to better understand the zoonotic risks they pose.